# Selected Insect Pests of Economic Importance to *Brassica oleracea*, Their Control Strategies and the Potential Threat to Environmental Pollution in Africa

**Nelson Mpumi** [1,*], **Revocatus S. Machunda** [1], **Kelvin M. Mtei** [1] and **Patrick A. Ndakidemi** [2]

[1] School of Materials, Energy, Water and Environmental Sciences, The Nelson Mandela African Institution of Science and Technology, P.O. Box 447, Arusha, Tanzania; revocatus.machunda@nm-aist.ac.tz (R.S.M.); kelvin.mtei@nm-aist.ac.tz (K.M.M.)

[2] School of Life Sciences and Bioengineering, The Nelson Mandela African Institution of Science and Technology, P.O. Box 447, Arusha, Tanzania; Patrick.ndakidemi@nm-aist.ac.tz

[*] Correspondence: nelsonm@nm-aist.ac.tz; Tel.: +255-755938619

**Abstract:** The most common destructive insect pests affecting cabbages in African smallholder farmers include *Plutella xylostella*, *Helula undalis*, *Pieris brassicae*, *Brevycoryne brassicae*, *Trichoplusia ni* and *Myzus persicae*. Those insect pests infest cabbages at different stages of growth, causing huge damage and resulting into huge yield losses. The African smallholder farmers use cultural and synthetic pesticides to control insect pests and minimize infestations. The cultural practices like crop rotation, weeding and handpicking are used to minimize the invasion of cabbage pests. However, those practices are not sufficiently enough to control cabbage insect pests although they are cheap and safe to the environment. Also, the African smallholder famers rely intensively on the application of broad-spectrum of synthetic pesticides to effectively control the cabbage pests in the field. Due to severe infestation of cabbages caused by those insects, most of African smallholder farmers decide to; first, increase the concentrations of synthetic pesticides beyond the recommended amount by manufacturers. Secondly, increase the rate of application of the synthetic pesticides throughout the growing season to effectively kill the most stubborn insect pests infesting cabbages (*Brassica oleracea* var. capitata). Thirdly, they mix more than two synthetic pesticides for the purpose of increasing the spectrum of killing the most stubborn insect pests in the field. All those scenarios intensify the environmental pollution especially soil and water pollution. Moreover, most of insecticides sprayed are made with broad-spectrum and are hazardous chemicals posing environmental pollution and threats to natural enemies' ecosystems. Therefore, this paper reviews *Brassica oleracea* var. capitata insect pests and control measures as a potential environmental pollution threat in African smallholder farmers.

**Keywords:** phytochemicals; *Plutella xylostella*; Helula undalis; biological control and cultural practices

## 1. Introduction

Cabbage, (*Brassica oleracea* var. capitata), is an essential leafy plant grown as an annual mainly for use as a vegetable crop [1,2]. Cabbage is a leafy vegetable of *Brassica* family and is round, oblate and pointed shapes. *Brassica* family vegetables are very important and are intensively grown for resource poor African smallholder farmers for subsistence and a source of income [3]. Cabbage is a water loving plants. Thus, it is grown in the areas with enough supply of water. Cabbage has soft, light green or whitish inner leaves covered with harder and dark green outer leaves. Cabbages are full of vitamins such as vitamin K and C and the dietary fibers and full of potassium and manganese [4] and it has antioxidant and anti-inflammatory properties [5] in the body of human being. Also, it has detoxifying

effect due to its high sulphur and vitamin C contents [6]. Cabbage is commonly used all over the world and can be prepared in a number of ways for eating and most frequently, it is included as either a cooked or raw part of many salads [1,7].

However, cabbage is susceptible to insect pest infestations in the field, which causes huge loses to the growers [1,7]. The major insect pests infesting cabbages in Africa include the diamondback moth (*Plutella xylostella*), the cabbage webworm (*Helula undalis*), cabbage white butterfly (*Pieris brassicae*), cabbage aphid (*Brevycoryne brassicae*), cabbage looper (*Trichoplusia ni*) and green peach aphids (*Myzus persicae*) [1,7]. However, diamondback moth, (*Plutella xylostella*), and cabbage webworm (*Helula undalis*) are the most destructive insect pests of brassica vegetable crops in many parts of African countries and are mostly damaging in the tropics and subtropics [8–10]. Those insect pests infest the *Brassica oleracea* crop at different stages of growth, causing huge destruction to the cabbage crop during the growth stages [1,8,11] and lastly results into huge yield losses in the field. Although, *B. oleracea* production in most African countries is practiced by a number of smallholder farmers, there is limited information on cabbage yield losses caused by insect pests in the field.

Damage and impact of damage on yield depends on the cabbage variety grown and other elements of the ecosystems like natural enemies and weather conditions, fertilizer and water availability [10,12]. Due to those constraints, the control and management of the cabbage insect pests and diseases is very poor in most African smallholder farmers [3]. In African countries, synthetic pesticide applications are mostly preferred and are excessively used frequently and above the labelled rates [13].

The smallholder famers in African countries rely intensively on the application of broad-spectrum synthetic pesticides to control the insect pest of cabbages [10,14]. Most pesticides (about 79%) used are broad-spectrum synthetic insecticides, including organophosphate (OP) (profenofos, WHO Class II), pyrethroid (cypermethrin and deltamethrin, WHO Class II) and avermectin-based formulations (abamectin, WHO Class Ib) in which WHO class Ib is categorized as Highly Hazardous and WHO class II is Moderately Hazardous [10,15]. The indiscriminate and intensive applications of synthetic insecticides reduce natural enemy ecosystems' biodiversity by causing the death of useful natural enemies such as spiders, lacewings and hoverflies [16,17], pollinators like bees, and butterflies which pollinate flowers [1,18,19] and lady bird beetles which are important and could help in the reduction of aphids [16]. Moreover, the massive synthetic pesticides spraying contribute to a serious environmental pollution especially water and soil pollution [15]. Spraying of incorrect synthetic pesticides, excessive application, incorrect timing, the incorrect mixing of chemicals and forged chemicals lead to insecticide resistance which causes farmers to spray even more pesticides [12,20] in order to control the insect pests and consequently resulting into environmental pollution especially water and soil pollution which endangers the aquatic and soil ecosystems [20] respectively.

However, botanical pesticides from pesticidal plants such as *Tephrosia vogelii* and *Azadirachta indica, Annona squamosa, Cupscum frutensces,* and *Allium sativa* could be the promising alternatives since they have been used to successfully control insect pests in cereal crops [21,22]. Mudzingwa et al. [3] indicated that, botanical pesticides are naturally occurring compounds derived from medicinal and pesticidal plants and they contain groups of active compounds of diverse chemical nature and have an average residual life of 2–5 days. However, there is little information regarding the application of botanical pesticides for the control of cabbage insect pests in the field. Table 1 indicates some of the pesticidal plants reported by some scholars to control cabbage insect pests in the field and in the greenhouse nursery.

**Table 1.** Some of the pesticidal plants reported to control cabbage insect pests in Africa.

| Chemical and Pesticidal Plant | Insect Pests Controlled | The Area of Study | Reference |
|---|---|---|---|
| Garlic (*Allium sativum* L.) and Hot Pepper (*Capsicum frutescens* L.) | *Brevicoryne brassicae* (L.), *Plutella xylostella* (L.), *Helula undalis* (Fab.) and *Trichoplusia ni* (Hub) | In a greenhouse nursery | [1,23] |
| *Lantana camara* (L.) and *Azadirachta indica* (A. Juss) | *Plutella xylostella*, *Brevicorynebrassicae* and *Hellula undalis* | In a greenhouse nursery | [7] |
| *Ageratum conyzoides, Chromolaena odorata Synedrella nodiflora, Capsicum frutescens, Nicotiana tabacum Cassia sophera, Jatropha curcas, Ricinus communis* and *Ocimum gratissimum* | Cabbage aphids (*Brevicoryne brassicae*) and diamondback moth (*Plutella xylostella*) | It was a field cage experiment | [24] |
| *Lantana camara* (L.), *Azadirachta indica* (A. Juss), *Capsicum annuum* (L.) and *Curcuma longa* (L.) | Diamondback Moth, *Plutella xylostella* L. (Lepidoptera: Plutellidae) | It was conducted at the field | [25] |
| Plant extract Neem azal - S | *Brevicoryne brassicae* and *Bemesia tabaci* | It was conducted at the field | [26] |
| *Tephrosia vogelii, Allium sativum* and *Solanum incanum* | *Brevicoryne brassicae* in *Brassica napus* done in greenhouse nursery | It was conducted in the greenhouse nursery | [3] |

## 2. The Biological Cycle of *Brassica* Species and Their Common Insect Pests in Africa

### 2.1. Propagation and Biological Cycle Length of Selected Brassica Species

The propagation and regeneration of *Brassica* species has been successful using seeds and different explants like petioles, cotyledons, stems and shoot tips [27] (Table 2). Shoot regeneration and rooting of *Brassica* species are successfully obtained from cotyledons and hypocotyl explants [28]. The shoot tip explants of *Brassica* species are reported to be effective for initiating shoots and roots [29]. Table 2 shows the *Brassica* species propagation and biological cycle length.

**Table 2.** Propagation and biological cycle length of *Brassica* species.

| Name of *Brassica* Species | Propagation | Biological CycleLength | References |
|---|---|---|---|
| *Brassica oleracea* L. | Conventional propagation is through seed, with seedlings being raised in beds or modules and then transplanted to field sites. However, some *B. oleracea* subspecies, such as tronchuda, can be propagated through vegetative from stem and side shoot cuttings whereby the stem and side shoot cuttings are obtained from 5-week old plants, which is rooted, and transplanted as normal cuttings | Seed germinates within 5 days after sowing at 20–25 °C. | [30] |
| *Brassica juncea* L. | Conventional propagation is through seeds also, it has been successful by using petioles, cotyledons, stems and shoot tips as explants. | Seed germinates within 5 days after sowing at 20–25 °C. | [27,31] |
| *Brassica napus* L. | Conventional propagation is through seeds. The seedlings are raised in seedling trays or in a seedbed. Also, it is propagated successful by using stems, cotyledons, nodal stems and hypocotyl as explants in vitro. | The seeds take 3–5 days to emerge at 20–25 °C | [27,32,33] |
| *Brassica rapa* L. | Conventional propagation is done using seeds but also, the propagation is successful through petioles, stems, cotyledons, stems and shoot tips as explants in vitro. | The seeds require 3–5 days to germinate at 20–25 °C | [27] |
| *Brassica campestris* L. | Conventional propagation is through seeds. Also, petiole and cotyledons can be used in the development of a plants in vitro culture. Four day seedlings are enough to give a viable *Brassica campestris* plants | The seeds require 3–5 days to germinate at 20–25 °C | [27,33,34] |
| *Brassica nigra* | The propagation is done using seeds. The small seeds require a level and a well-prepared seedbed. | The first leaves are usually visible within 48 h | [35] |

*2.2. The Common Insect Pests Affecting Cabbages in Africa*

Many insect pests such as diamondback moth (*Plutella xylostella*), cabbage webworm (*Helula undalis*), cabbage white butterfly (*Pieris brassicae*), the cabbage aphids (*Brevycoryne brassicae*), green peach aphids (*Myzus persicae*) and cabbage loopers (*Trichloplusia ni*) [1,7] (Table 3) hinder the proper cabbage crop production on the field in Africa. Those insect pests (Table 3) infest the cabbage crops at different stages of growth, causing significant damage to the crop [11] and resulting into huge cabbage yield losses. Krishnamoorthy [36] showed that, cabbage insect pests all together can cause 52% yield loss on cabbage. Severe infestation by *Plutella xylostella* usually causes huge economic crop losses and may result in 100% yield loss of the *Brassica oleracea* [37]. Due to heavy infestations which result into huge losses, the African smallholder farmers of cabbages spray four or more than four times in a month and two or more than two mixed insecticides into the field for strongly and effectively control of the cabbage insect pests [38,39]. The consequence of that scenario is environmental pollution especially water and soil, detrimental effects to non-target organisms and endanger the health of the human being [20]. This section reviews the major insect pests of economic importance infesting cabbage crop at different stages of growth in African countries and how the control measures are potential water and soil pollution threat.

**Table 3.** Common insect pests of cabbages [1,7].

| Common Name | Scientific Name | Parts of Cabbages Damaged |
|---|---|---|
| Dimondback moth | *Plutella xylostella* | Cabbage heads and foliar tissues |
| Cabbage webworm | *Helula undalis* | Leaves, petioles and heads of cabbages |
| Cabbage white butterfly | *Pieris brassicae* | Head of cabbage and leaves |
| Cabbage aphid | *Brevycoryne brassicae* | Tips, flowers and leaves |
| Green peach aphids | *Myzus persicae* | Tips, flowers, developing pods and leaves |
| Cabbage looper | *Trichloplusia ni* | Leaves, stems and veins of leaves |

2.2.1. Cabbage Looper (*Trichoplusia ni*)

The cabbage looper (Figure 1A) (*Trichoplusia ni*) is a moth found in the family noctuidae a family which is commonly referred to as owlet moths [40]. Its common name comes from its preferred host plants and distinctive crawling behavior. The members of noctuidae are brown or gray night-flying moths whereby the larvae infest the growth of cruciferous vegetables [41]. Cruciferous vegetables like cabbages, bok choy and broccoli are the main host plants to cabbage lopper and hence, the reference to cabbage in its common name [42]. The larvae is called a looper since it arches its back into a loop when it crawls [40]. While crucifers are preferred, however, over 160 plants can serve as hosts of cabbage looper larvae [40]. The adult cabbage looper is a migratory moth and its migratory behavior can be found in a wide range of host plants and this contribute to its wide range of distribution [41].

The cabbage looper larvae is a vegetable pest for crucifers and has been reported to damage broccoli, cabbages, cauliflowers, chinese cabbages, collards, kale, mustards, radish, turnip and watercress [41]. The cabbage looper larvae interfere with plant growth and marketability by making irregular holes of variable shapes (Figure 1A,B) while feeding on the leaves of the host cabbage plants [23]. Although it is not extremely destructive, but it is becoming difficult to control and manage due to its broad distribution and resistance to many insecticides [23,42]. Therefore, African smallholder farmers rely intensively on the application of synthetic pesticides to control the cabbage insect pests. However, synthetic pesticides result into environmental pollution, insect pest resistance and contaminate the foods which consequently threaten the human health [23]. Therefore, environmental benign, the botanical pesticides from *T. vogelii, S. aromaticum* and *C. dichogamus* can be utilized to control the insect pests in the field instead of synthetic pesticides [24]. Although the potentialities are ignored, but botanical pesticides have been in use for centuries by smallholder farmers in developing countries to control insect pests of both field and stored products [25,43]. Therefore, they could be used to control cabbage insect pests in the field to minimize the infestation.

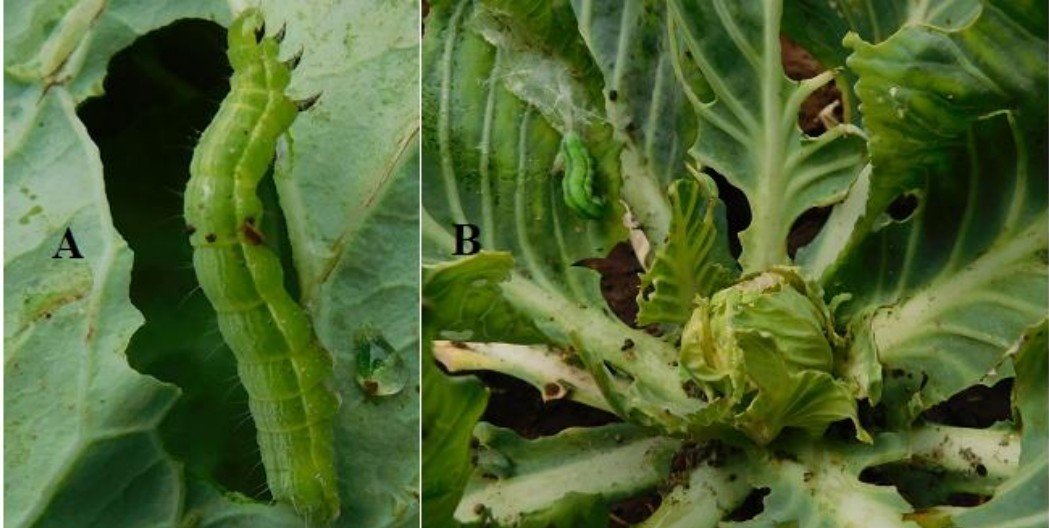

**Figure 1.** (**A**) Mature larva of the cabbage looper, *Trichoplusia ni*. (**B**) The cabbage plant damaged by Cabbage looper larva. Photograph by Nelson Mpumi, NM-AIST-Arusha, Tanzania.

### 2.2.2. Cabbage Webworm (*Hellula undalis*)

Among the most destructive insect pests which attack cruciferous vegetables is the cabbage webworm (Figure 2A) (*Hellula undalis*) (Lepidoptera: Pyralidae) [44]. The cabbage webworm (*Hellula undalis*) is a major pest of cruciferous crops in the tropics and subtropics [45]. It is a widespread species in the world especially in Europe across Asia to the Pacific and also, in African countries [46,47]. Shine et al. [48] reported that *H. undalis* is distributed mostly in tropical and subtropical regions but can similarly be found in countries with moderate climates.

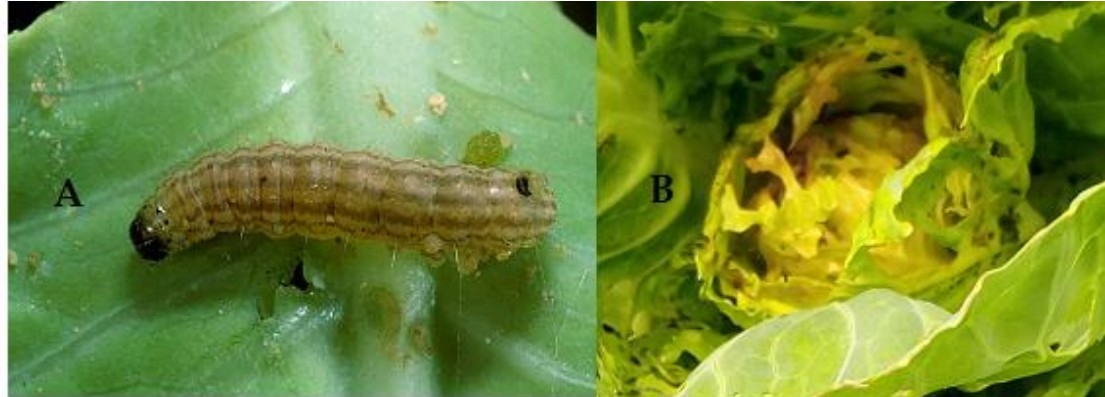

**Figure 2.** (**A**) Mature larva of the cabbage webworm, *Hellula undalis*. Photograph by Lyle Buss, Entomology and Nematology Department, University of Florida (March, 2016). (**B**) The cabbage plant head damaged by larva of *H. undalis*. Photograph by Nelson Mpumi, NM-AIST Arusha, Tanzania.

Ebenebe et al. [46] reported that, *Helulla. undalis* larva causes a serious and severe damage to the leaves and the heads of cabbages (Figure 2B) in the field. According to Waterhouse and Norris [49] *H. undalis* feeds on a variety of plants especially the Brassicaceae family members. Waterhouse and Norris [49] revealed that, *H. undalis* larva can cause a huge yield loses of up to 100% to crucifers crops in the field and its management is not well taken into account. The larvae feed on leaves, petioles, growing points and stems [47]. According to Sivapragasam and Aziz [44] and Waterhouse and Sands [47], the plants in which *H. undalis* larvae feed include broccoli, head cabbage, chinese cabbage, spoon cabbage, daikon radish, horseradish, mustard, radish and turnip. Shine et al. [48] revealed that, *H.*

*undalis* is a very serious agricultural pest to crucifer crops grown by the African smallholder famers. The incidence of *H. undalis* did not depend on the number of insecticide applications, but depend highly to host crop abundance and the temperature of the area [10]. The larvae make mines in the leaves and bore into the stem and later, they tunnel into the heart of the plant, destroying the bud causing the leaves to become distorted and stunted [1]. A study done by Sivapragasam and Aziz [44] indicated that, a single larva of cabbage webworm, can either cause a number of deaths to the young plant or lead to the formation of unmarketable multiple heads on relatively older plant. On the field, a low population of larvae can cause very huge significant losses to the cabbage crop and in untreated cabbages, losses could go as high as 99% [46]. Although, the larva can be present throughout the cropping season, it is severe only during the period between transplanting and the heading stage of cabbage [50].

Currently, African smallholder farmers rely intensively on the application synthetic pesticides as the only effective control method to the cabbage webworm on the field [44]. The effective insecticides which are used to control cabbage webworm worldwide and Africa particularly, include permethrin, abamectin, teflubenzuron, chlorfluazuron, triflumuron, phenthoate, exthofenprox and Lamda-cyhalothrin and among those insecticides, abamectin is found to be the most effective of the other insecticides [44]. However, some are reported hazardous and therefore unwise and overuse of those insecticides can result into severe environmental pollution especially water and soil, development of insect resistance to some of insecticides and health problems to human being. Therefore, there is a need of searching and utilizing benign and environmental friendly botanicals from pesticidal plants as cabbage insect pests control strategy.

### 2.2.3. Diamondback month (*Plutella xyllostela*)

The diamondback moth (Figure 3A) (*Plutella xylostella*), sometimes called the cabbage moth, is a moth species belonging to the family Plutellidae and genus *Plutella* [50,51]. Badenes-Perez et al. [52] reported that *Plutella xylostella* is believed to have originated in Europe, South Africa, or the Mediterranean region, but it has now spread worldwide. The diamondback moth is the dominant and most destructive insect pest of crucifer crops worldwide [25]. Justus and Mitchell [53] reported that, *Plutella xylostella* larva feeds on the leaves between the large veins and midribs of cruciferous crops and the plants which produce glucosinolates. *Plutella xylostella* larva prefers to feed on the lower leaf surface, leaving the upper epidermis intact creating a "window-paning" effect (Figure 3B) [25]. Timbilla and Nyarko [11] showed that, severe feeding damage (Figure 3B) stunts and destroys the cabbage heads and can cause heads to abort leading to huge yield depression and total crop loss. The most cabbage plant damage is caused by larval feeding resulting in a complete removal of foliar tissues and disrupt head formation in cabbages, broccoli and cauliflower [25]. The destruction of cruciferous crops by diamondback moth larva reduces the quality and the marketability of the cabbage crops and hence yield losses due to *P. xylostella* can go up to 100% [12,37] and vary widely depending on the season and severity of pest infestation [54].

Generally, it is estimated that, diamondback moth causes an annual loss of about 16 million dollar on the basis of 2.5 per cent damage even on the protected crop [55]. Also, in the tropics, diamondback moth causes threat of great loss of 90% and above to crucifer production crop [56]. Therefore, there is a need to conduct a research to determine the cabbage losses due to infestation of diamondback moth in various parts of Africa.

The diamondback moth and its larvae control in cabbage by African smallholder farmers is still deeply dependent on chemical insecticides although their use is connected with many adverse and lethal consequences. Inappropriate and excessive application of chemical insecticides result into environmental pollution especially water and soil pollution [57,58]. Pedigo and Rice [59] indicated that, extreme use of insecticides also induces resistance development in target pests as well as killing beneficial organisms like pollinators such as bees and other natural enemies such as spiders, lacewings and ladybird beetle. Therefore, the benign, environmental friendly botanicals have to be searched

to control this pest instead of relying on the synthetic pesticides which have many negative impacts and problems to the environment. The benign and environmental friendly control measures with broad spectrum of the activities are the botanicals (phytochemicals), the chemicals from pesticidal plants [24]. Those alternatives with antifeedant, repellency and insect growth regulators of their natural origin having non-neurotoxic modes of action to human being and low environmental persistence can be applied.

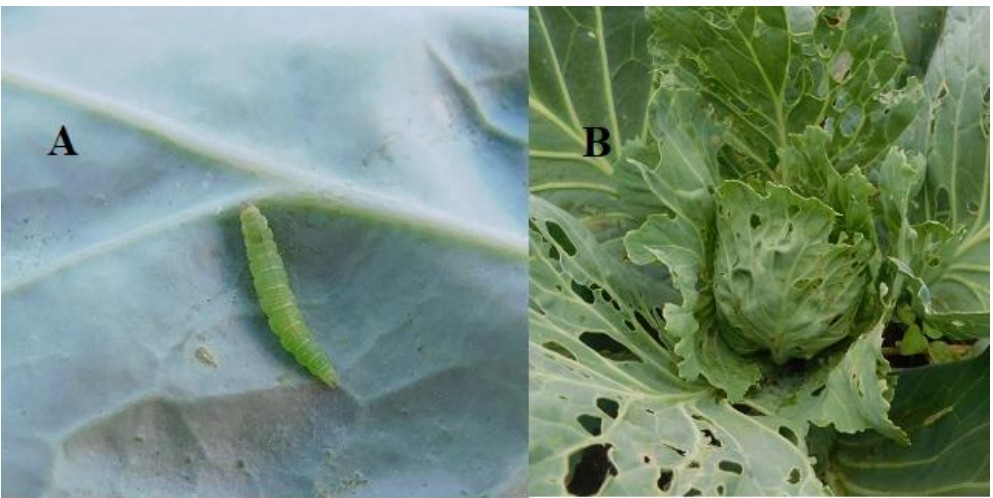

**Figure 3.** (**A**) Mature larva of the Diamond back moth, *Plutella xylostella*. (**B**) The cabbage plant damaged by larva of *P. xylostella*. Photograph by Nelson Mpumi. NM-AIST, Arusha, Tanzania.

Botanical pesticides are not only effective against crop pests but remain safe to the environment and to natural enemies [60]. In developing countries, botanicals have been in use for centuries by smallholder farmers to control insect pests both in field and storage [25]. For instance nicotine, rotenone and pyrethrum were famous and among the botanical insecticides used in those days [43. Those chemicals from pesticidal plants possess one or more useful properties like repellency, anti-feeding, fast knock down, flushing action, biodegradability, broad spectrum of activity and ability to reduce insect resistance [43,61]. Therefore, there is a need to use the environmental friendly products for instance, the botanicals/phytochemicals from *Tephrosia vogelii*, *Syzygium aromaticum* and *Croton dichogamus* to control cabbage insect pests in the field.

### 2.2.4. The Cabbage Aphids, (*Brevicoryne brassicae*)

Cabbage aphid (Figure 4A) (*Brevicoryne brassicae*) belongs to the family Aphididae of the order Hemiptera [62] and the genus *Brevicoryne* [63]. The name is derived from two Latin words "brevi" and "coryne" and which means "small pipes" [62]. In those aphids, there are two small pipes called cornicles or siphunculi at the posterior end which can be observed when using hand lens during the observation [64]. The cornicles of the cabbage aphid are comparatively shorter than the cornicles of other aphids except those of the turnip aphid, *Lipaphis erysimi* [64]. The short cornicles and the waxy coating present on cabbage aphids differentiate cabbage aphids from other aphids which can attack the same host plants [64,65]. The cabbage aphid is native to Europe, but now has a world wide distribution [65,66] and can be found in Africa, Asia, Canada, Australia [67], America, India, China and Netherland [63] and also in African countries.

Jahan et al. [68] and Moharramipour et al. [69] indicated that, cabbage aphids are serious plant sap sucking pests worldwide. Those aphids are the most common damaging species causing significant yield loss to many crops of Brassicaceae, like the mustards and crucifers [3,69]. Blackman and Eastop [70] insisted that, cabbage aphids mostly attack growing parts of the host plants such as tips, flowers, developing pods, leaves and eventually cover the whole plants (Figure 4B) at high population.

According to Elwakil and Mossler [71] and Lashkari et al. [72] cabbage aphids (Figure 4A) have direct and indirect damaging effects to cabbage crops. The direct damage caused by this pest is by sucking cell sap, secrete honey dew which result into sooty mold formation on leaves and shoots and indirect damaging effect is as a vector of 20 plant viral diseases in a wide range of plants. According to Valenzuela and Hoffmann [73], the damaging viruses transmitted by cabbage aphids are such as potato leafroll virus, potyviruses, beet western yellows, beet yellows, cauliflower mosaic, cucumber mosaic, lettuce mosaic, turnip mosaic and watermelon mosaic. High population and feeding of cabbage aphids result into curling, distortion and yellowing of leaves, stunting plant growth, deforming developing heads, damaging of flowers and green pods and discoloration of any growth stage and part of plants [64,67]. Feeding by cabbage aphids can stop terminal growth resulting into reduced plant size and yield [74].

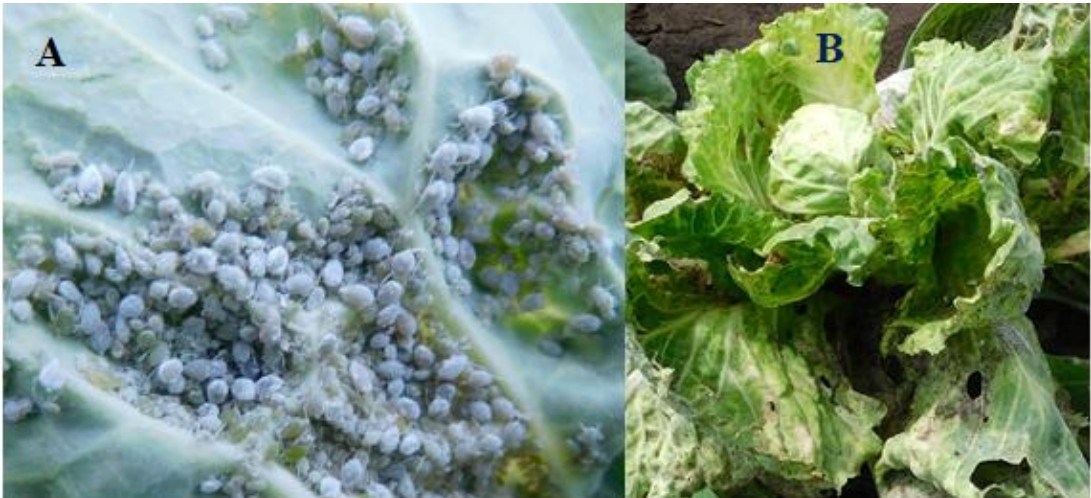

**Figure 4.** (**A**) Cabbage aphids, *Brevicoryne brassicae*. (**B**) The damaged plant cabbage by cabbage Aphids. Photograph by Nelson Mpumi. NM-AIST, Arusha, Tanzania.

Eliminating weeds in Brassicaceae field borders is one of the cultural methods which may help to reduce the population and damaging of the cabbage aphids [72,75]. However, cultural methods alone are less effective to completely control the cabbage aphids from the farmers' field [62]. So, biological control can play a major role in the natural suppression of aphids. Among the biological controls which can be applied to control the aphids are the natural enemies such as ladybird beetles adult and larvae, lacewing larvae, syrphid fly larvae, predatory bugs and lacewing larvae [72,76]. Other biological control agents are entomopathogenic fungi, which particularly can be applied during the periods of high humidity and precipitation [72,75]. However, natural enemies alone and other biological controls are also insufficient to prevent economic damage by a rapidly increasing population of cabbage aphids [26].

Due to high pest pressure and damaging caused by those aphids on cabbages in African countries, growers resort to excessive and intensive chemical pesticides application for aphids and other insect pest management [62,76]. Chemical pesticides are intensively, excessively and doubly rated for insect pest management [62,76]. However, intensive and heavily reliance on the application of the synthetic pesticides results into extreme soil and water pollution and pose serious threats to the non-target organisms including human beings [62]. For instance, Bami [77] reported that, every year, one million people are suffering from pesticide poisoning in India. The pesticides poisoning threatens the health of human being and the natural enemies. Also, the soil pollution threatened the soil ecosystem. Decomposers are also in danger due to soil pollution through excessive and intensive application of synthetic pesticides [20].

Due to those problems associated with the application of synthetic pesticides, there is a need of assessing the potential of botanical pesticides from various plants such as *T. vogelli, S. aromaticum and C. dichogamus* for cabbage aphid control and management in the field. Botanicals from different pesticidal plants have many advantages over synthetic pesticides such as local availability and inexpensive pest control agents [16,62].

### 2.2.5. The Green Peach Aphids (*Myzus persicae*)

The green peach aphid, (*Myzus persicae*) (Figure 5A), is found throughout the world and can be present at any time throughout the year [78]. Generally, its color is pale green, and there are two forms of green peach aphids; winged and wingless forms [79]. The green peach aphid have prominent cornicles on the abdomen that are markedly swollen and club like in appearance [70]. The frontal tubercles at the base of the antennae are very prominent and are convergent [79]. Winged forms of the green peach aphid have a distinct dark patch near the tip of the abdomen; wingless forms lack this dark patch [80]. The green peach aphid is adapted to high environmental temperatures [78].

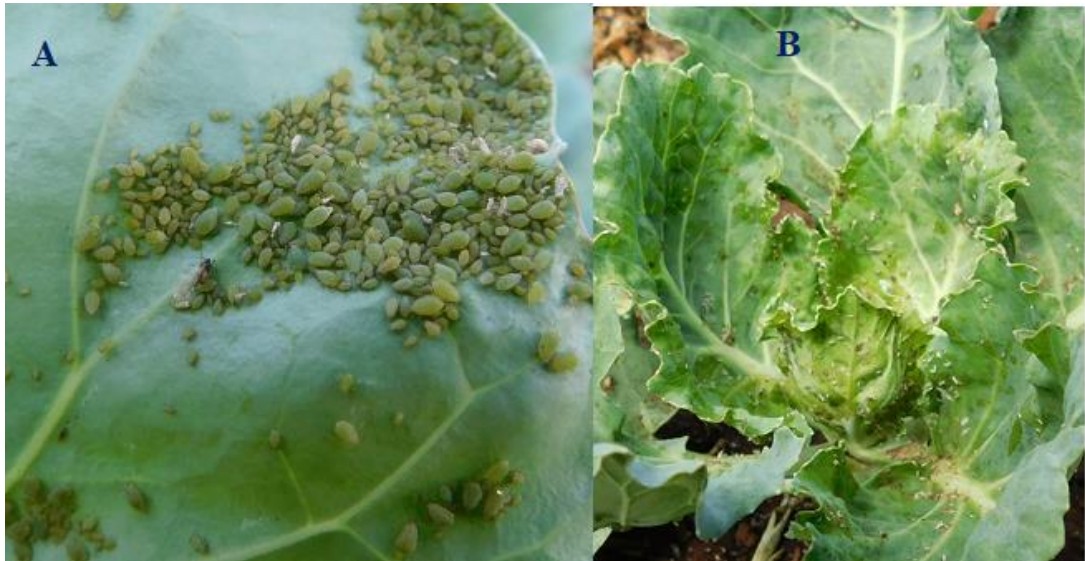

**Figure 5.** (**A**) Green peach aphids, *Myzus persicae*. (**B**) The cabbage affected by green peach aphids. Photograph by Nelson Mpumi. NM-AIST, Arusha, Tanzania.

Blackman and Eastop [70] and Gu et al. [78] showed that, over 40 plant families are hosts of green peach aphids. According to them, those plants include woody and herbaceous plants including vegetable crops in the family Solanaceae, Chenopodiaceae, Compositaceae, Brassicaceae, and Cucurbitaceae. Some of the host plants which support the growth and development of green peach aphids include cabbages, spinach, asparagus, bean, beets, broccoli, Brussels sprouts, carrot, cauliflower, cantaloupe, celery, corn, cucumber, fennel, kale, turnip, eggplant, lettuce, mustard, okra, parsley, parsnip, pea, pepper, potato, radish, squash, tomato, turnip, watercress and watermelon [70]. Moreover, Gu et al. [78] added that, many flower crops and ornamental plants are also suitable for growth and development of green peach aphids. Those all crops differ in their vulnerability to green peach aphids, but the actively growing plants and plants' parts, or the youngest plant tissues often are affected by large aphid populations [80]. Broadleaf vegetables are particularly very suitable host plants for green peach aphids. Therefore, the broadleaf vegetables create pest infestation problems in nearby crops [78]. The green peach aphids can achieve very high densities on young plant tissues, causing water stress, wilting and reduced growth rate of the plant [79].

Anstead et al. [81] and Umina et al. [80] indicated that, adults and nymphs of aphids can damage the crops in three ways: - firstly, they feed directly on young tender plant tissues and causes drying out

of shoots, wilting and distortions of the plants' parts (Figure 5B); secondly, they produce honeydew which falls onto foliage and becomes blackened by sooty mould fungi; and thirdly, they spread more than 100 viruses. According to Anstead et al. [81], de Little and Umina [82] and Valenzuela and Hoffmann [83], the damaging viruses transmitted by green peach aphids are such as potato leafroll, potyviruses in pepper, beet western yellows, beet yellows, cauliflower mosaic, cucumber mosaic, lettuce mosaic, papaya ringspot, turnip mosaic and watermelon mosaic. These viruses affect the proper growth and development of the crops and reduce the marketability. The damaging levels caused by green peach aphids are characterized by large numbers of aphids found on the underside of leaves sucking the plant saps [81,82]. In addition to attacking plants in the field, the green peach aphid can readily infest vegetables and ornamental plants grown in glasshouses [78]. Umina et al. [80] reported that, the aphids feed by sucking sap from leaves and flower buds, but the entire crop foliage may be covered when populations are large resulting in reduced or stunted growth of young plants. The extensive feeding of green peach aphids on crops enforces plants to turn yellow and the leaves to curl downward and inward from the edges resulting into wilting, stunted growth and finally death of the crops [82]. When young plants are infested in glasshouses and then transplanted into the field, the fields will not only be inoculated with aphids but insecticide resistance may be introduced [78]. de Little and Umina [82] insisted that, the green peach aphid is considered the most important vector of plant viruses in the world. Also, contamination of harvestable plant material with aphids, or aphid honeydew, causes the loss of the food quality and quantity [83]. Therefore, prolonged aphid infestation of crops can reduce the yield of crop products.

The green peach aphid is attacked by a number of common predators such as lacewings, lady beetles, syrphid flies and parasites, including the parasitic wasps (*Lysiphlebus testaceipes, Aphidius matricariae, Aphelinus semiflavus,* and *Diaeretiella rapae*), and is susceptible to the fungus disease, *Entomophthora* spp. All those natural enemies together with field sanitation helps to control the incidence and spread of viruses transmitted by green peach aphid, but it does little to control the aphid itself. So, the smallholder farmers rely on the application of chemical insecticides to control the green peach aphids in the field. The use of chemical pesticides to control *M. persicae* on the food crops is increasing globally [84]. For instance in African countries like Tanzania, *M. persicae* are now extensively controlled with insecticides in oilseeds, pulses, and vegetable crops [79]. However, heavy reliance on insecticides to manage aphid populations result into strong insect pest resistance and *M. persicae* has probably developed resistance to more insecticides than any other insect species [84,85]. Therefore, broad spectrum insect pest control strategies are needed to ensure the aphids are controlled.

The severe damaged caused by insect pests in various parts of *B. oleracea* (Table 4 and Figure 5) compel the African smallholder farmers to increases the doses of the synthetic pesticides during the application.

**Table 4.** The parts of *Brassica oleracea* damaged by insect pests, signs and their effects [1,12,19,61].

| Insect Pests | Parts of Cabbages Damaged | Signs of the Damaged Crop | Effects |
|---|---|---|---|
| *Plutella xylostella* | Cabbage heads and remove foliar tissues | Stunts and destroys the cabbage heads | Reduces quality and marketability of cabbage crops |
| *Helula undalis* | Leaves, petioles and heads of cabbages | Distorted of plant organ and stunted growth | Deaths to young plants and formation of unmarketable multiple heads |
| *Pieris brassicae* | Head of cabbage and leaves | Deforming developing heads of cabbage and leaves | Interfere with plant growth and marketability of the cabbages |
| *Brevycoryne brassicae* | Tips, flowers and leaves | Curling, distortion and yellowing of leaves, stunting growth, deforming developing heads | Stop terminal growth leading to reduced plant size and yield |
| *Myzus persicae* | Tips, flowers, developing pods and leaves | Yellowing of leaves, stunting growth, deforming developing heads and curling of leaves | Wilting, stunted growth and finally death of the crops |
| *Trichloplusia ni* | Leaves, stems and veins of leaves | Large irregular holes of variable shapes on the leaves | Interfere with crop growth and marketability of the cabbages |

### 2.2.6. *Brassica oleracea* Insect Pests with Insecticides' Resistance

Some of the important pests of *Brassica oleracea* such as the Diamond Back Moth (*Plutella xylostella*), cabbage webworms (*Hellula undalis*), whiteflies (*Bemicia tabaci*) and aphids (*Brevicoryne brassicae* and *Myzus persicae*) have developed resistance to a wide range of commonly used pesticides [86]. For instance, *Plutella xylostella* is documented to have developed resistance to a number of insecticides [87]. The tests done in four regions in New Zeland between 1999 and 2000 reported that, *P. xylostella* developed resistance to synthetic pyrethroids [88]. The resistance of *P. xylostella* to pyrethroids is based on the oxidative detoxification of monooxygenase enzymes [89]. The level of resistance of *P. xylostella* to cypermethrin can be 1096 fold [90]. However, the resistance of *P. xylostella* to pyrethroids insecticides can be even 2880 fold [87]. Varma and Sandhu [91] reported the resistance of *P. xylostella* to DDT and parathion organochlorine insecticides in India. Also, *P. xylostella* is reported to developed resistance to fenitrothion and malathion [92], cypermethrin, decamethrin and quinalphos [93], cartap hydrochloride, diafenthiuron and flufenexuron [87,92]. The major reasons for *P. xylostella* to develop resistance to insecticides includes: the increase in number of sprays, misuse of pesticides, inappropriate dosages used by farmers and frequency of applications [94]. Apart from the insecticides resistance developed by *P. xylostella*, also cabbage webworms (*Hellula undalis*), whiteflies (*Bemicia tabaci*), aphids (*Brevicoryne brassicae* and *Myzus persicae*) have developed resistance to cypermethrin, decamethrin, chlorpyrfors, malathion and lamda-cyhalothrine [88]. Therefore, *Brassica oleracea* insect pest resistance to synthetic pesticides calls for search of the alternatives products which can effectively control those insect pests in the field.

## 3. The Biological Life Cycle and Common Practices Used to Control Cabbage Insect Pests

### 3.1. The Biological Life Cycle of Brassica Species Insect Pests

For the proper integrated management and effectively control of the common insect pests of *Brassica* species, there is a need to at least understand briefly the biological life cycle of them. Table 5 briefly present generation number, the eggs per adult and the biological length of selected common insect pests of *Brassica* species.

**Table 5.** Generation number, eggs/adult and the biological length of selected insect pests of *Brassica* species.

| Cabbage Insect Pests | Generation Number | Eggs/Adult | Length of Biological Cycle | Reference |
|---|---|---|---|---|
| *Plutella xylostella* | It complete 13–14 generations annually. | 187 eggs per adult during the life time in *Brassica oleracea* var. capitata | It requires 19.4 days to complete the life cycle. | [95] |
| *Hellula unalis* | It ranges from 7–8 generations annually. | 175 eggs per adult during her life time in *Brassica* species. | The total time for the life cycle ranged from 22.75 days at 35 °C to 89.93 days at 20 °C and that depend on the hosts | [96] |
| *Trichoplusia ni* | At least one generation can be completed per month successfully under favorable weather conditions. | 300 to 600 eggs are produced by a female during her life time. | It requires 18 to 25 days when are held at 32 to 21 °C, respectively to complete the life cycle. | [97,98] |
| *Brevycoryne brassicae* | An average of 15 generations are completed during the crop seasons. | A female can give birth 30–50 nymphs without mating and the colony will consists of females only. When mating occur, a female can lay 5–7 eggs. The colony will consists of males and females | It ranges from 16–50 depending on temperatures. It is shorter at high temperatures and long at low temperatures | [76,99] |
| *Myzus persicae* | The maximum number of generation is 20 and 21 in a year and it depends on favourable weather conditions | The oviparous female can oviposit four to thirteen eggs. The viviparous female can give birth to a mean fecundity of 75 offspring. | The length mean of reproductive period is 20 days. | [100] |

### 3.2. Common Practices Used to Control Cabbage Insect Pests

Insect pest infestation in Cabbages (*Brassica oleracea*) is a very serious problem because they cause huge damage which result into huge cabbage loses in the field. They cause a huge, severe and serious damage to the leaves and the heads of cabbages [46] resulting very huge yield loses. The yield loses can reach up to 100% of crucifers crops in the field [47]. This part reviews the common practices (Table 6) used by African smallholder farmers to efficiently and effectively control the insect pests of the *Brassica oleracea* in the field.

**Table 6.** Cabbage insect pest control practices done by African smallholder farmers.

| Practices | Advantages | Disadvantages |
|---|---|---|
| Cultural practices Biological practices | They are cheap and safe to the environment. Affordable by most of smallholder farmers. Have little effect to the populations of beneficial insects. Have low human toxicity and little environmental pollution problems. | Those methods are effective when used in Combination with other practices Requires enough expertise, enough skills and knowledge in developing them and apply for the control of cabbage insect pests. |
| Chemical pesticides | Fast effective, reliable against a wide range of insect pests and easily tested for new insect pests | Causes environmental pollution, threatened human health, kills none target organisms, and destroy the Ecosystems |
| Botanicals | Less persistence in the environment, harmless to none target organisms, low mammalian toxicity, rapid in action | It is not easy to standardize the extracts, rapid degradation and affected by weather conditions |

### 3.2.1. Chemical Pesticides

Synthetic pesticides have been used intensively for many years to control crop insect pests. Alavanja [101] reported that, about 5.6 billion pounds of synthetic pesticides are used to protect foods and commercial crops. In Africa, the predominant pesticide groups used to control the insect pests of crops and cabbages include insecticides mainly organophosphates, fungicides and herbicides [15,45]. The effective insecticides which are used to control cabbage insect pests worldwide and Africa particularly, include permethrin, abamectin, teflubenzuron, chlorfluazuron, triflumuron, phenthoate, exthofenprox and Lamda-cyhalothrin and among those insecticides, abamectin is found to be the most effective of the other insecticides [44]. According to Labou et al. [10], carbaryl, methomyl, permethrin, and trichlorfon are effective in controlling larvae of insect pests in the field. Also, Nyirenda et al. [102] reported that, synthetic pesticides are effective, reliable against a wide range of insect pests and act quick and easily tested for new insect pests. Moreover, some of synthetic pesticides such as DDT is used in public health programs and commercial applications, for lawn and garden applications and in around the homes [103] despite its toxic effect and persistence nature in the environment. Cypermethion, carbaryl and λ-cyhalothrin (karate) are used to control the pests [104] in crops. Therefore, the use of synthetic pesticides assisted to significantly reduce crop losses and improve the yield of crops such as grains crops, leafy vegetables and potatoes [105].

However, besides their beneficial effects, most of synthetic pesticides such as endosulfan, lindane and DDT have potential environmental pollution and public health impacts [15,105]. The reports by De Bon et al. [106] and Weinberger and Srinivasan [12] indicated that, many synthetic pesticides used are persistent in the environment, threaten the human health, kills non-target organisms and destroy the ecosystems (Figure 6). Moreover, in the environment, the application of synthetic pesticides commonly results into water and soil contaminations (Figure 6), development of insect resistance to the pesticides applied and threatened the food security for human being [105,107,108]. Apart from that, also, the availability of synthetic pesticides in distant rural areas where *Brassica oleracea* smallholder farmers are living and practice cabbage cultivation are either unreliable or are expensive. Again, synthetic pesticides are extremely diluted to ineffective concentrations by dishonest traders and they are toxic to non-target insect pests (Figure 6) [109]. When synthetic pesticides contaminate the soil, the soil ecosystems are threatened. Synthetic pesticides have high persistence in the environment [43]

which means the pesticide can stay in the soil for many days, even years and can cause bioaccumulation and biomagnifications in the bodies of the organisms in the environment. In soil, organisms are killed and can result into biomagnification and bioaccumulation [45,101]. Biomagnification refers to increase in the concentration of a pollutant such as a pesticide toxic chemical in the tissues of tolerant organisms from one trophic level to another trophic level [110]. This increase can occur as a result of; first persistence of that chemical substance whereby the substance cannot be broken down by environmental processes into simple and less harmful substances, secondly food chain energetics in which the substance's concentration increases progressively as it moves up a food chain and lastly, low or non-existent rate of internal degradation or excretion of the substance which is often due to water insolubility [110]. Bioaccumulation refers to the increase in the concentration of a substance in certain tissues of organisms' bodies due to absorption of the chemical substance from food and the environment [110]. The contamination of soil affect soil macro and microorganisms which are decomposers. The soil pollution affects the quality of soil chemically, biologically and physically and therefore reduces the soil fertility and productivity. Apart from that, also, synthetic pesticides affect the health of farmers during preparation, application in the farms and the consumption of cabbages. Therefore, botanical pesticides which are affordable and have health benefits to the applicators, consumers and the environment should be used for the control of insect pests of the crops [21].

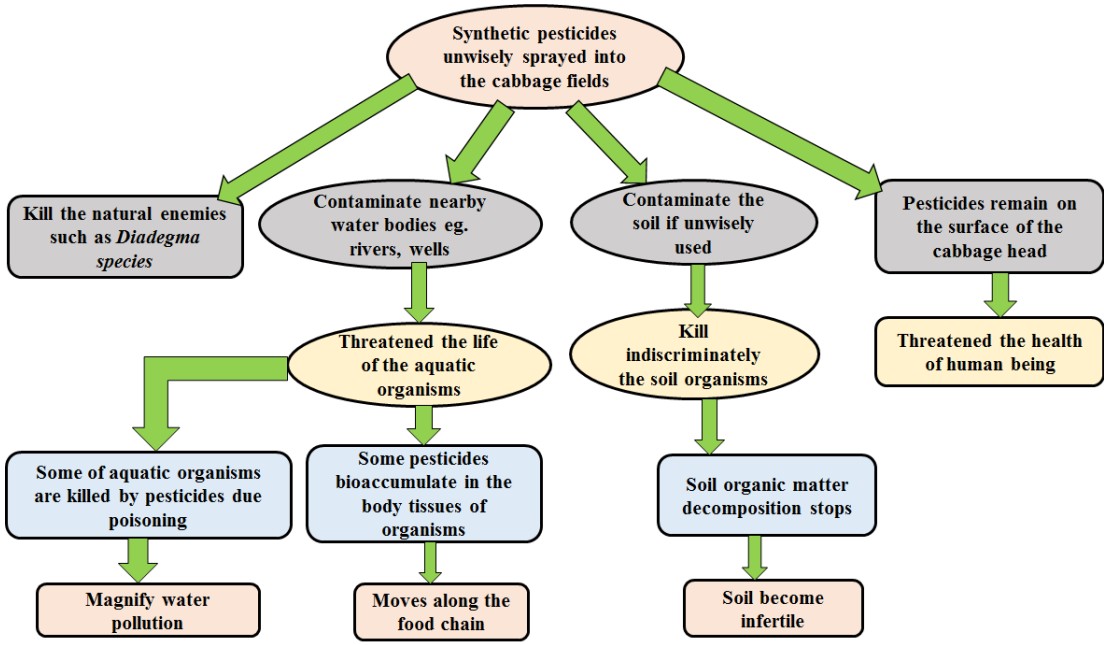

**Figure 6.** The summary of the fate of synthetic pesticides when applied heavily on *Brassica oleracea*.

### 3.2.2. Cultural Methods

Cultural control practices refers to a broad set of management techniques which are used to minimize or eliminate insect pests by agricultural producers to achieve the crop production goals. There are several cultural practices, which African smallholder farmers use to reduce infestations of the insect pests of the crops in the field [111]. Generally, the cultural practices such as site selection, intercropping practices, crop rotation, seed selection and sowing date can minimize the invasion of insect pests in the crops [112]. Weinberger and Srinivasan [12] reported that, when the intercropping or trap crops are grown along with the crucifers in the same field, pest populations are kept at low. Apart from that, cabbage looper, (*Trichoplusia ni*) can be managed by crop rotation when lettuce is introduced into the garden after *Brassica oleracea* to eliminate it. Also, using clean planting materials and transplanting only healthy and vigorous insect-free seedlings, reduce the infestation of *Brassica*

*oleracea* insect pests in the field. Moreover, uprooting and burning of any remains of cabbage and other related plant debris, protecting seedling beds and either using greenhouses with close mesh nets or screens also, reduce infestations by insect pests.

The planting time is important to be observed since proper planting time of *Brassica oleracea* in the field minimizes the infestation of insect pests. For instance, aphid infestations in *Brassica oleracea* is reduced by early sowing time [7]. Therefore, sowing time can affect the population of aphids and other arthropods attacking cabbages. Moreover, Weinberger and Srinivasan [12] reported that, mechanical means such as weeding, or natural methods can be used to control insect pests of cabbages in the field. In other agronomic studies, row spacing and plant density and weed control are used to control insect pests of cabbage production in the field. Moreover, removing weeds, aid to the control aphids. Handpicking, minimizes large pests such as slugs, leatherjackets or caterpillars. However, this method is quite efficient especially in a small garden. Apart from that, cultural control methods are friend to the environment and to the health of human being, less costly in terms of money and time, minimizes chances for biotype selection and also not harmful to non-target organisms. However, most of those cultural practices alone are not sufficiently enough to protect cabbage insect pests in the field although they are cheap and safe to the environment [2].

### 3.2.3. Biological Control

The term "biological control" has been used long time ago to describe either the use of live predatory insects, entomopathogenic nematodes, microbial insecticides and natural enemies or the use of the natural products extracted or fermented from numerous sources to suppress populations of different insects pests [113]. In cabbage production, biological control is involved in the control of cabbage insect pests. For example, microbial insecticides are involved in cabbage looper management and their potential role has so far been fully realized. Gupta and Dikshit [114] reported that, the most widely known microbial pesticides are varieties of the bacterium *Bacillus thuringiensis* (Bt) which control certain insects in cabbage, potatoes and other crops. *Bacillus thuringiensis* has been used for long time to effectively suppress the cabbage looper and has little effect to the populations of beneficial insects [113]. *Bacillus thuringiensis* produces a protein that is harmful to specific insect pests of cabbage like diamond back moth and cabbage looper which when the protein is ingested by either pest or pest larvae, the mid gut of the pest is damaged, eventually killing it [114]. Generally, *Bacillus thuringiensis*, controls certain caterpillars, beetles and flies [115] and have low human toxicity and little environmental pollution problems. The natural enemies such as predators, parasitoids and pathogens are involved in the management of cabbage insect pests [111]. The predators such as spiders, lacewings, lady beetles, ground beetles, rove beetles, hover flies, and true bugs [16,116] attack, kill and feed on insect pests affecting the production of crops. Also, those organisms can kill and feed into the insect pests which affect the production of leafy vegetables in the field. Ladybird beetles, family *Coccinelidea*, both adults and larvae feeds on aphids [117] and as a consequence reduce the populations of aphids in the cabbage field. Ladybird beetles are stronger, larger and normally are more intelligent than the prey [111] and hence attack several hosts in a short period of time. Parasitoids such as many species of wasps and some flies parasitize and kill other invertebrates [111]. Some of species of parasitoids, when are in immature stage develops on or within a single insect host forming mummies and finally kill the host [118]. Parasitoids are parasitic when are in immature stage and kill their hosts as they reach maturity [116]. Biological control is safe and eco-friendly, and therefore, more study is required for insect pests control in cabbages [119]. Despite the safety, lower environmental and low toxicity effect of biological control methods, African smallholder farmers have little knowledge and little understanding on the application of biological control method for cabbage insect pest management and therefore rely much on the use of synthetic pesticides to control cabbage insect pests. Relying on application of synthetic pesticides for cabbage insect pests control lead to environmental pollution. Therefore, application of safe, environmental friendly, the phytochemicals from pesticidal plants for cabbage

insect pests control should be employed to minimize the environmental pollution from synthetic pesticides (Figure 7).

### 3.2.4. Botanical Pesticides

Botanical pesticides have been used as alternatives to synthetic insecticides to control cereal crop insect pests in the field and in the storage because they pose little threat to the environment, to ecosystems and to human health [21,43]. In the middle of the 17th century, pyrethrum, nicotine and rotenone were recognized as effective insect control agents [43,120,121]. In fact, Arannilewa et al. [122] revealed that, many plants with medicinal properties demonstrated potential insect pests control agents. The plants with pesticidal properties, comprises numerous active chemicals, which affects the reproductive and digestive process of a number of important pests [43,114]. Le Roy and Wrana [123] reported that, the plants with active chemical compounds of medicinal and pesticidal properties are botanicals. Botanical insecticides like tobacco extracts, neem oil extracts have been found useful for pests control in cereal crops. In addition, the plants like *Tephrosia vogelii*, *Azadirachta indica*, *Annona squamosa*, *Cupscum frutensces*, *Allium sativa* have potentials for controlling cereal crop insect pests [22], *Aristolochia ringens* and *Alium sativum* displayed antifeedants, contact poisons and repellents properties against *Sitophilus zeamais* [122] and *Nicotiana tabacum*, *Azadirachta indica*, *Eucalyptus camaldulemsis* and *Swietenia mehagani*, indicated effectiveness against aphids [122]. Therefore, those potentials of botanical pesticides as mentioned above can be employed in controlling the cabbage insect pests in the field. The mixture of bioactive compounds in plants have potential advantages in terms of efficacy and short life span development of resistance [115]. Those chemicals have little mammalian toxicity, degrade rapidly in sunlight, air, and moisture and therefore, are less persistence in the environment, and are rapid in action to the insect pests [24,120]. Moreover, botanicals have little effects to non-target organisms and natural enemies of insect pests (Figure 7), have little or no toxic effect on plant growth and cooking quality of the edible part of the crop and also, are less expensive and easily available in the farmers' natural environment (Figure 7) [21]. However, botanical pesticides are extensively used for the protection of cereal crops from insect pests in the field and storage and there are very little information available on the use of botanical pesticides to control cabbage insect pests in the field.

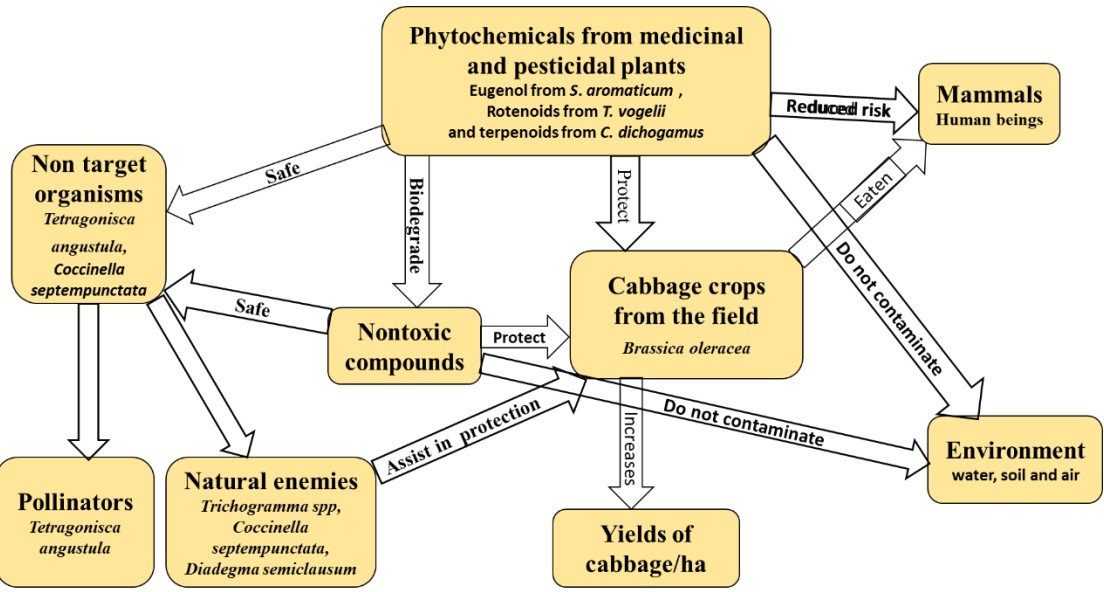

**Figure 7.** The summary of the advantages of botanicals/phytochemicals when applied on cabbages.

#### 4. The Fate of Pesticides Used to Control Cabbage Insect Pests in African Smallholder Farmers

Generally, pest control in cabbage by smallholder farmers is still heavily dependent on chemical insecticides although their use is associated with many undesirable and sometimes lethal consequences [54]. The herbicides such as triazines (atrazine, simazine, terbuthylazine, propazine, cyanazine, terbutryn, prometryn), phenylureas (diuron, linuron, isoproturon, chlortoluron) and anilides (alachlor, acetochlor, metolachlor), insecticides such as organophosphorus (malathion, chlorfenvinphos, dimethoate, parathion-methyl, azinphos-ethyl, chlorpyrifos, fenitrothion) and organochlorine (lindane and DDTs) [15] and some of their metabolites are the most common pesticides found in the soil, the surface and groundwater bodies [45]. The pollution of the groundwater and surface water by unwise use of pesticides in agriculture threatened the soil organisms and their ecosystems (Figure 6) and drinking water resources [10,45]. Moreover, excessive use of insecticides also induces resistance development in target pests as well as killing beneficial organisms such as pollinators for example bees and natural enemies [16,37,59] in the field.

For instance, due to the severe infestation of cabbage caused by insect pests, most of smallholder farmers in African countries decide to effectively suppress and kill the insect pests of cabbages in the field through the following ways. Firstly, they increase the concentration of the synthetic pesticides during the application in the cabbage field [105]. That means, the smallholder farmers use the synthetic pesticides beyond the recommended amount by the manufacturers which result into the extreme soil pollution. Secondly, they increase the rate of application of the synthetic pesticides to the field. Sometimes pesticides is applied twice in a week to strongly and effectively kill the very stubborn insect pests of cabbages. For instance, Ngowi et al. [39] and Ntow et al. [105] reported that, 5 to 16 times pesticide applications per crop is practiced for the whole growing season in African countries, with onion being the most treated crop, followed by tomatoes and cabbages being the last and the frequency bases on a weekly application in many situations. For instance, Orr and Ritchie [124] reported that, the farmers spray on average of 19 times in tomatoes and 14 times in cabbages in Malawi throughout the growing season. Also, Ahouangninou et al. [38] revealed that, 70% of vegetable growers in Southern Benin apply four to five times chemical treatments per month while doubling or tripling the recommended dosage. According to De Bon et al. [106], the smallholder farmers believe that, the frequency of pesticides applications certainly prevent the insect pests attack effectively. The improper and overuse of synthetic pesticides magnify water and soil pollution which finally threatened the water and soil ecosystems [45]. Thirdly, they mix more than two synthetic pesticides at the same time in order to increase the spectrum of destroying and killing various insect pests in the field. In so doing, the environment is threatened [45]. Specifically, water bodies and soil pollution occur due to intensive application of synthetic pesticides without considering the recommendation of the manufacturers. The synthetic pesticides kill the organisms in the environment indiscriminately [24,125] which imply that, both beneficial and harmful organisms are killed indiscriminately. The decomposers, fishes and other organisms in water for example are usually affected by extreme application of the synthetic insecticides meaning that, the decomposition of organic matter to release nutrients into the soil is affected. As a result the soil become infertile and less productive. In water bodies, aquatic organisms such as fishes, anglerfishes, sponges, shrimps, phytoplankton and zooplanktons are killed and can result into biomagnification and bioaccumulation [24,125]. The increase of toxic substance along the food chain can occur as a result of first, persistence of that chemical substance in the environmental media whereby the substance cannot be broken down by environmental processes into simple and less harmful substances; secondly the food chain energetics in which the substance's concentration increases progressively as it moves up a food chain, and; lastly, low or non-existent rate of internal degradation or excretion of the substance which is often due to water insolubility [110].

#### 5. Conclusions

The continuous use of synthetic insecticides to control cabbage insect pests should be highly discouraged or minimized because they contribute to environmental pollution and health risks to

consumers. The chemicals from pesticidal plants, should be encouraged for cabbage insect pest control because they are less persistence in the environment, have low risks to the consumers and less food contamination alongside the easy way of obtaining and preparation. Also, the botanicals do not affect the natural enemies' ecosystems when used and therefore, can function with various components, including natural enemies, cultural control and biological control for the control of cabbage insect pests. As a matter of fact, the use of botanicals to control cabbage insect pests should be encouraged and supported to African smallholder farmers to reduce pollution of the environment through the use of synthetic pesticides. This review insist on the ecological research with a focus on use of botanicals for the cabbage insect pest control in the field as available knowledge on the ecology of most invertebrate pests attacking cabbage crops in Africa is limited.

**Author Contributions:** All authors have read and agreed to the published version of the manuscript.

**Funding:** This work was supported by African Development Bank (AfDB) through The Nelson Mandela African Institution of Science and Technology (NM-AIST), Arusha, Tanzania.

**Acknowledgments:** The authors grateful to The Nelson Mandela African Institution of Science and Technology, Arusha (NM-AIST) Tanzania for providing institutional support for the ongoing study and African Development Bank (AfDB) for funding the study. Also, I thank the Tanzania Agricultural Research Institute (TARI), Tengeru, Arusha, Tanzania and Wazazi Association in Boro Village in Kilimajaro region, Tanzania for providing land for establishing the study plots. We thank the technicians of the institutions for field assistance and data collection.

**Conflicts of Interest:** The authors declare no conflicts of interest regarding the publication of this paper.

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
