# Peer review of "Selected Insect Pests of Economic Importance to Brassica oleracea, Their Control Strategies and the Potential Threat to Environmental Pollution in Africa"

_sustainability, doi:10.3390/su12093824_

Round 1
Reviewer 1 Report
- Abstract: Enter the botanical name of cabbage in parenthesis while mentioning it for the first time on line 13
- Table 1 & 3: Italicize the name of the species
- Incomplete sentences at different points e.g., line 68
- Redundant arguments e.g. line 76-82 and 128-132
- No references provided in Table 3 and 4
Author Response
Please see our responses
|
The reviewer’s comments |
The responses from the Authors |
|
The botanical name of Cabbage (Brassica oleracea var. capitata) was entered in parenthesis on line 13 |
|
The name of species were italicized in Table 1 & 3. |
|
· Incomplete sentences at different points e.g., line 68 |
After passing through the work again, the incomplete sentences e.g., line 68 were completed. |
|
The redundant arguments for example line 76-82 and 128-132 were corrected |
|
Table 3 and 4 are our own compilation based on the literatures |
Reviewer 2 Report
Thanks for the opportunity to review this manuscript about the insect pests of Brassica vegetables and their control strategies. In general the article needs revision. The presentation lacks clarity/consistency. I provide below a few suggestions that, if the authors decide to implement into the paper, the paper will improved.
- In several instances authors reported that synthetic insecticides caused environmental pollution. This should be supported with results of published studies (e.g. examples of insecticides that found in water etc.). If all the insecticides causes environmental pollution why registered in most of the countries. It is not correct to generalize that pesticides caused pollution if some instances this happens due to inappropriate application.
- The authors reported the insecticides that control the insect that infect Brassica vegetables in both sections 2 and 3. This information should be presented only in section 3.
- In section 2 authors should present information about biological cycle for each Brassica species (e.g. propagation, biological cycle length etc.).
- In section 3.4., the authors reported that botanical insecticides can used for insect control in vegetables. Authors should cited results about the efficacy of the botanical insecticides against Brassica or other crop pests.
- In lines 491-493 the authors reported that botanical pesticides are used extensively for the protection of cereals crops. This should be supported with references. The authors should add information about the efficacy of certain botanical pesticides against specific insects. Also, the authors should support the statement that botanical pesticides are used extensively.
- Lines 551-553: In this phrase authors reported that synthetic pesticides are harmful for human health and causes environmental polltuion. Why then the synthetic pesticides are registered for insects control around the world? Usually the pollution problems are related with inappropriate application of insecticides.
- Authors should add a paragraph about the Brassica pests with insecticides resistance around the world in order to support the fact that control of these pests should be based not only on the application of pesticides.
- The section “3.2. cultural practices” needs to be further analyzed. The authors should give more examples with results of other studies.
Reviewer 3 Report
Evaluation
Authors present a comprehensive overview of the literature on the insect pest of cabbage (Brassica oleracea), common practices used to control insect pests and potential environmental threat in Africa due to excessive use of synthetic pesticides by smallholder farmers. The manuscript is properly organized, however, needs a scope of improvement (listed below) before publishing it:
- Line 43: “Manganese” should be written in a small letter
- Lines 65-69: the following sentence seems to be unfinished – please reformulate it: “Most of treatments (about 65 79%) sprayed are made with broad-spectrum synthetic insecticides, including organophosphate (OP) (profenofos, WHO Class II), pyrethroid (cypermethrin and deltamethrin, WHO Class II) and avermectin-based formulations (abamectin, WHO Class Ib) in which (Ib = Highly hazardous; II = Moderately hazardous)”
- Lines 78-79: the content in both lines is similar – please rewrite
- Line 94: I suggest the following numbering: “2.”
- Lines 112 and 143: the chapter title should not be in bold
- Lines 163-169: the same content (repetition) in lines 163-164 and 168-169 – please check and correct.
- Line 248: please change “damaging effetcs” to “damaging effects”
- Lines 344-346: the following sentence is unclear/unfinished – please rephrase: “The common signs of the damaged parts of B. oleracea crops and their effects caused by insect pests infestation which due to their severe effects (Table 3), compel the African smallholder farmers to overly and increases the doses of the synthetic pesticides during the application into fields.”
- Lines 351-352: please provide references in Table 3
- Line 355: the numbering should be “3.”
- Line 381: it should be “De Bon et al. [82] and Weinberger and Srinivasan [12]”
- Line 381: it should be “synthetic pesticides used are persistent”
- Line 409: please change “pesticdes” to “pesticides” and provide reference to the Figure 6 or indicate that it is your own compilation
- Line 495 and 501: please provide references in the Table 4 and Figure 7 or clearly indicate that it is your own compilation based on the literature
- Line 502: the chapter numbering should be “4.”
- Line 512: there should be “Figure 6” instead “Figure 1”
- Line 550: instead “5.0” should be “5.”
- Line 551: please remove one word “synthetic”
Author Response
Please seethe attachment

Round 2
Reviewer 1 Report
Thanks for making the appropriate corrections to make the manuscript more interesting to the readers.
Table 4: If the table is a compilation of data from different works of literature, it deserves to be cited.
Reviewer 2 Report
The authors following the comments during the reviewing process improved the manuscript.
*A table with information about biological cycles (e.g. generations number, propagation dynamic (e.g. eggs/adult), and length of biological cycle) of the main brassica insects should be added.
